# Effect of a Single Multi-Vitamin and Mineral Supplement on Nutritional Intake in Korean Elderly: Korean National Health and Nutrition Examination Survey 2018–2020

**DOI:** 10.3390/nu15071561

**Published:** 2023-03-23

**Authors:** Hyoeun Kim, Seung Guk Park

**Affiliations:** Department of Family Medicine, Inje University Haeundae Paik Hospital, Busan 48108, Republic of Korea

**Keywords:** dietary supplements, vitamins, minerals, aged, nutrition surveys

## Abstract

Inadequate nutritional intake is common, especially among elderly individuals. Although micronutrient intake may help fill nutritional gaps, the effects of multi-vitamin and mineral supplements (MVMS) among the Korean elderly are not well known. Therefore, we investigated the nutrition-improving effects of a single MVMS. A total of 2478 people aged ≥65 years who participated in the Korea National Health and Nutrition Survey 2018–2020 were analyzed. Nutrient intake from food and supplements was measured using the 24 h recall method. We compared the nutritional intake and insufficiency between the food-only group (*n* = 2170) and the food and MVMS group (*n* = 308). We also evaluated the differences in inadequate nutritional intake after taking MVMS with food. The analysis included vitamins A and C, thiamine, riboflavin, niacin, calcium, iron, and phosphorus. The proportion of insufficient intake ranged from 6.2% to 80.5% for men and from 21.2% to 82.4% for women, depending on the nutrients. Intake of MVMS with food was associated with lower rates of inadequacy (3.8–68.5% for men and 3.3–75.5% for women) compared to the food-only group. The results suggest that micronutrient deficiency frequently occurs in the Korean elderly population and can be improved by MVMS intake.

## 1. Introduction

For many older adults, food remains the sole source of nutrients. As a result, older adults in communities and hospitals have been shown to be at a high risk of malnutrition in assessments using the Mini Nutritional Assessment (MNA), a widespread nutritional screening and assessment tool. The prevalence of malnutrition risk in this population is relatively high, ranging from 19% to 27% [1,2]. The Korean Frailty and Aging Cohort Study (KFACS) showed that 14.3% of community-dwelling older adults were malnourished [3]. The increase in the aging population suggests that the number of malnourished older people will also increase, causing severe social and health problems.

Older people experience nutritional deficiencies for various reasons, including anorexia, oral health problems, biological changes in the digestive system, chronic diseases, multiple drug use, and social factors [4,5,6]. Nutritional deficiencies in food intake can also lead to macronutrient and micronutrient deficiencies [4]. In particular, micronutrient deficiencies such as iron and vitamins A, C, D, and E are more common in older adults than in younger adults [7]. Micronutrient deficiencies can adversely affect various aspects of the health of elderly individuals, including immune function [8,9], frailty [10], osteoporosis [11], and longevity [12], through multiple pathways related to cell differentiation, oxidative stress, muscle and bone metabolism, inflammation, and decreased immunity [10,13,14]. Recent studies have also suggested that micronutrient deficiencies, such as vitamins, can cause abnormal brain functions such as oxidative stress, mitochondrial dysfunction, and neurodegeneration, leading to various neurological disorders such as Alzheimer’s disease, Parkinson’s disease, and depression [14]. An increase in dementia or other neurological diseases can be another burden in an aging population.

Previous studies [7,15,16,17,18,19,20,21,22] have shown that taking vitamins and minerals through dietary supplements (DS) can help improve micronutrient deficiencies and achieve the recommended levels of these micronutrients. In Korea, some studies have shown that the supplementation of vitamins and minerals through DS in addition to food can help reach the recommended daily nutritional intake for the general population [17,18] and adolescents [22]. Some studies conducted in Korea showed the DS usage patterns and micronutrient deficiencies in elderly Korean individuals [23], and some studies suggested the improvement effect after taking multi-vitamin and mineral supplements (MVMS) in older people. Therefore, the improvements in older Korean individuals taking MVMS with food need to be clarified.

We hypothesized that the intakes of micronutrients in older Koreans need to be increased to meet the recommended nutritional intake and that the MVMS may help improve nutritional status. In order to exclude overlapping effects due to the combination with other DS and to determine the effect of only a single MVMS, this study was conducted with subjects who only took MVMS purely by excluding all multi-users. To address these hypotheses, in this study, based on data from the large nationally representative Korean National Nutrition and Health Survey (KNHANES) 2018–2020, we investigated the micronutrient intake status in older Korean people. We also evaluated the MVMS-related degree of reduction in the proportion of insufficient nutritional intake below the estimated average requirement (EAR) by comparing the findings for the group consuming food and a single MVMS with that consuming food alone.

## 2. Materials and Methods

### 2.1. Study Population

This cross-sectional study used data from KNHANES 2018–2020. The KNHANES provides nationally representative data on Koreans’ health status, health awareness and behavior, and food and nutrition status. The three surveys—Health Interview, Health Screening, and Nutrition Survey—were conducted through face-to-face interviews and self-administered questionnaires. The nutrition survey evaluated dietary habits, including DS use, food security, and food and nutrient intake during the previous day, which were obtained by a well-trained nutritionist using a 24 h recall method. Of the 23,471 people who participated in the 2018–2020 survey, 5110 were over 65 years old. Of these 5110 older adults, 643 participants were excluded from the study because they did not complete the nutrition survey. Among the remaining 4467 participants, 243 who did not answer the DS-related questionnaire were excluded from the study, and 115 participants with missing weight values were also excluded. Two thousand one hundred and seventy people who had not taken a DS during the previous 24 h were assigned to the food-only group (*n* = 2170). Of the 1939 individuals who took a DS, 768 responded that they were taking MVMS. After excluding 460 individuals who took an additional DS along with MVMS, 308 individuals were finally assigned to the “food + MVMS group” (*n* = 308) (Figure 1). The Institutional Review Board of the Korea Centers for Disease Control and Prevention approved the KNHANES 18-20 data (IRB Approval Number: 2018-01-03-P-A, 2018-01-03-C-A, and 2018-01-03-2C-A).

### 2.2. Baseline Characteristics

Sociodemographic factors, including age, sex, education level, household income level, smoking status, frequency of alcohol consumption, physical activity level, body mass index (BMI), and history of disease or cancer diagnosis, were surveyed using a self-administered questionnaire. BMI was calculated from the measured height and weight. Individuals were divided into two age groups aged 65–74 years and ≥75 years. Education levels were divided into elementary school or lower, graduation from middle school, high school, and university or higher. Household income was divided into high, middle-high, middle-low, and low based on household income quartiles. Smoking status was categorized as non-smokers, ex-smokers, and current smokers. The frequency of alcohol consumption was divided into none, less than once a week, two to three times a week, and more than four times a week. Physical activity level was divided into the following groups: moderate-intensity physical activity for more than 2 h 30 min each week, high-intensity physical activity for more than 1 h 15 min each week, combined moderate-intensity and high-intensity physical activity, and no moderate- or high-intensity physical activity. BMI was classified as the underweight range < 18.5 kg/m^2^, normal range between 18.5 kg/m^2^ and 23 kg/m^2^ (18.5 kg/m^2^ ≤ BMI < 23 kg/m^2^), overweight for values between 23 kg/m^2^ and 25 kg/m^2^ (23 kg/m^2^ ≤ BMI < 25 kg/m^2^), and obese for values ≥25 kg/m^2^. If the participants answered that they had been diagnosed with a disease or cancer by a doctor, they were recorded as having a disease or cancer. Diseases included hypertension, hyperlipidemia, cardiovascular disease, stroke, and diabetes mellitus. Cancers included stomach, colon, liver, breast, cervical, lung, and other cancers.

### 2.3. Definition of MVMS Users

DS included MVMS and vitamin C, omega-3 fatty acids, probiotics, red ginseng, calcium, vitamin A or lutein, propolis, vitamin D, iron, and other vitamin and mineral supplements. We defined the MVMS user group as those who only took MVMS among DS. The participants who answered “yes” to the question “Did you take a dietary supplement the day before the survey?” were defined as DS users. The brand name of the DS and the intake amount were confirmed by a nutritionist.

### 2.4. Vitamin and Mineral Intake from Foods and Supplements

A well-educated nutritionist visited the examiner’s home and obtained information on dietary habits and food frequency through a 24 h recall method and a semi-quantitative food frequency questionnaire. Using the DS survey, we determined product names and ingredient amounts, intake duration, dose, frequency, and whether the DS was taken one day before the survey. We calculated the daily intake of nutrients, from total energy, carbohydrates, protein, fat, fiber, vitamin A, thiamine, riboflavin, niacin, vitamin C, calcium, iron, and phosphorus, from food and the MVMS. Nutrient intake through DS that was evaluated in this study included intake of calcium, phosphorus, iron, vitamin A, vitamin B1, vitamin B2, niacin, and vitamin C, since the KNHANES presented only the intake of the above micronutrients among the components included in MVMS. The percentage of intake below the EAR and upper limit (UL) compared with the nutritional intake standard was calculated [24]. The EAR and UL standard values for each nutrient were applied according to the sex and age of the participant. EAR was evaluated for all eight nutrients, and UL was only analyzed for nutrients with recommended baseline values: vitamin A, vitamin C, calcium, iron, and phosphorus. In the Dietary Reference Intakes for Koreans 2020 [24], the UL of niacin was divided into limits for nicotinic acid and nicotinamide. Thus, niacin was excluded from the analysis because it did not reflect the total niacin levels.

### 2.5. Statistical Analysis

Statistical analysis was conducted using STATA 15.0 SE (Stata Corp., College Station, TX, USA), with a significance level of *p* < 0.05. The 2018–2020 KNHANES is a multistage stratified cluster with combined data. Therefore, we applied the weights presented for each survey year. Categorical variables were expressed as percentages and standard error (SE), and χ^2^ analysis was used to compare the two groups. The intake of nutrients was compared between the “food-only” group and the “food + MVMS” group using multivariate regression analysis after adjusting for variables such as age, sex, education level, household income, smoking status, frequency of drinking, physical activity, BMI, and presence of diseases and cancer. The degree of intake below the EAR and the changes after adding MVMS were evaluated as percentage SE.

## 3. Results

In this study, the intake rate of DS among elderly individuals aged 65 years or older who participated in the survey on DS was 49.9% (males, 45.3%; females, 53.4%). Among participants taking DS, 39.5% (males, 40.5%; females, 38.9%) took MVMS, and 16.7% (males, 19.2%; females, 15.2%) took MVMS alone.

Table 1 shows the basic characteristics of the two groups. In assessments of sociodemographic variables, the “food + MVMS group” included a high proportion of individuals aged 65 to 74 years and a low proportion of those with low household income. The two groups showed no differences in sex or education level. In assessments of health-related variables, the food + MVMS group included a lower proportion of current smokers and a higher proportion of individuals performing a moderate-to-high-intensity exercise. The two groups showed no difference in alcohol consumption frequency and no difference in terms of disease-related variables such as BMI, chronic disease diagnosis, and cancer diagnosis.

In a comparison of the daily intake of nutrients through food alone in the two groups, males showed a higher intake of total calories, fat, riboflavin, and calcium in the food + MVMS group than in the food-only group, while females showed higher fat intake in the food + MVMS group than in the food-only group. The food + MVMS group showed a significantly higher intake of all eight nutrients in males and females than the food-only group, except for phosphorus in females (Table 2 and Table 3).

Figure 2 shows the proportion of people who consumed less than the EAR for each nutrient in the MVMS user group. We also evaluated the percentage differences in nutritional intake status after taking MVMS (Appendix A). When participants consume food only, more than 50% of male participants consumed less than the EAR of vitamin A (80.3%), vitamin C (80.5%), and calcium (67.3%), and more than 50% of female participants consumed less than the EAR of vitamin A (78.1%), niacin (69.4%), vitamin C (82.4%), and calcium (80.5%). In addition, the rates of inadequate intake of thiamine (21.0%), phosphorus (6.2%), and iron (18.2%) in males and iron (21.2%) in females were low, at less than 20%. After MVMS intake (in the food + MVMS group), micronutrient intake in older adults increased for almost all nutrients. The most significant improvement was observed in vitamin C levels in both sexes (68.5% in males and 75.5% in females). Additionally, niacin, riboflavin, thiamine, and vitamin A also showed improved intake rates lower than the EAR between 21.7% and 46.1% in both males and females (Appendix A). However, even after taking MVMS with food, more than 50% of the male and female participants did not reach the EAR for vitamin A and calcium intake.

Table 4 shows the percentage of people who exceeded the ULs for vitamins A and C and calcium, phosphorus, and iron. Some male participants exceeded the UL for vitamin A and iron with food intake alone. However, these cases were relatively few, and the female participants exceeded the UL for none of the nutrients by food intake alone. The nutrients consumed above the UL in the MVMS + food group were vitamin A and iron in both men and women; the corresponding ratios were 1.5% for vitamin A and 2.1% for iron in men, and 1.4% for vitamin A and 1.2% for iron in women.

## 4. Discussion

This study confirmed the effects of supplementing insufficient micronutrient intake with a single MVMS. The proportion of participants who did not achieve adequate micronutrient intake through their diet was greater than 50% for vitamin A, vitamin C, and calcium in men and vitamins A, niacin, vitamin C, and calcium in women. On the other hand, MVMS intake helped reduce the proportion of undernourished men and women. However, although MVMS intake improved the nutrient levels, more than 50% of the participants still required more vitamin A and calcium intake. Nevertheless, concerns about the problems caused by excessive intake were not serious.

We found that the “food + MVMS group” participants were relatively younger, suggesting that even among elderly individuals, older age is associated with a more vulnerable nutritional status. Other studies have also suggested that elderly individuals aged 70–75 years or older show lower intakes of most nutrients than those aged 65 years and older [7,25]. As the population ages, individuals become more vulnerable to malnutrition requiring more attention. We also observed that MVMS users had a healthier lifestyle. Previous studies have shown that supplement users are more likely to be female, educated, non-smokers, physically active, and knowledgeable about nutrition and health [17,23]. When we compared the nutrient intake obtained from food between the two groups of participants in this study, the intake of calories, fat, riboflavin, and calcium was higher in male MVMS users than in non-users. In contrast, among women, there was no significant difference in nutrient intake from food, except for fat intake. Consumption of DS implies a greater interest in health, and these individuals are believed to seek a higher intake of vitamins and minerals from high-quality meals. These differences were more pronounced with increased MVMS intake.

Inadequate vitamin and mineral intake in the elderly has already been described in previous studies [4,7,13,15,16,23,25,26], and the findings of this study were consistent with the previously reported data. A meta-analysis using various nutritional evaluations of the elderly in 37 Western countries showed that more than 50% of older adults did not receive sufficient amounts of thiamine, riboflavin, vitamin D, calcium, magnesium, and selenium from their diet [4].

In studies of nationally representative data from various countries, such as Canada [13], the United States [7,15], China [25], Korea [23], and the Philippines [26], which evaluated the amount of nutrients consumed using the 24 h recall method in the elderly, the extent of deficiencies were found to be particularly serious in Asian countries. Deficiencies in various nutrients, such as vitamins A [23,25,26], C [23,25,26], D [4,7,13], and E [7,15], thiamin [4,23,25,26], riboflavin [23,25,26], niacin [23], pyridoxine [25,26], cobalamin [25], folate [15,25,26], calcium [4,7,13,23,25,26], magnesium [4,7,13,15,25], zinc [25], selenium [25], iron [23,26], and phosphorus [23] have been reported previously.

Among the nutrients, vitamin A, vitamin C, and calcium showed a high inadequate nutritional ratio in our study. For vitamin A, approximately 80% of older Korean adults consumed less than the EAR, and even after taking MVMS, more than 50% still consumed insufficient amounts of vitamin A. Vitamin A is associated with visual function, immune function for the prevention of some viral infections in relation to host susceptibility [9], cell growth, and development, and bone health [27,28]. For vitamin A, the unit of calculation was changed from retinol equivalent (RE) to retinol activity equivalent (RAE) in KNHANES VII (2016) [29]. Due to the unit change, the measured carotenoid activity was lower, and vitamin A intake was reduced from before, even if the same amount was consumed, which exaggerated the deficiency. In the traditional Korean diet that is mainly based on rice, which is the primary diet consumed by the elderly, calorie intake was low and the intake of vitamins A and C as well as riboflavin, calcium, and iron were also low [30]. Therefore, Koreans who obtain vitamin A mainly through plant foods such as carotenoids appear to have lower intakes.

Overall, MVMS supplementation improved micronutrient intake in older adults. This improvement effect was mainly observed for vitamins. Although vitamin C showed one of the highest rates of insufficiency among participants in the food-only group, this vitamin also showed the best improvement after taking MVMS. The proportion of men and women who consumed vitamin C below the EAR was 80% in the food-only group, and the intake level was very vulnerable. The intake of fruits and vegetables, which can be considered representative vitamin C sources for the elderly, has tended to increase gradually in comparison with the past. However, consumption exceeding the recommended amount is still essential [31]. Vitamin C is a representative antioxidant that plays a role in enhancing immunity [8]. The intake of MVMS improved the insufficiency rate by nearly 70%. As a water-soluble vitamin, vitamin C has fewer known toxicities than fat-soluble vitamins and minerals; therefore, supplemental intake contributes to reducing inadequate intake in the elderly by providing these nutrients in a safer form.

Proper calcium intake reduces the risk of osteoporosis and fractures in the elderly [29]. As the fracture rate increases with age, osteoporosis is recognized as a significant disease for women and men. The findings indicated a severe calcium intake problem due to a marked lack of calcium intake in both individuals who consumed food alone and those who consumed an MVMS. Our results indicate the importance of a high-quality diet including calcium sources such as vegetables, milk, and fish for the elderly and additional single-nutrient calcium supplementation, if necessary.

In this study, the proportion of participants who exceeded the intake limit after taking MVMS was approximately 2% for vitamin A and calcium, which was lower than that in previous studies [15,16]. Excessive intake of vitamin A can lead to hip fractures, and excessive iron intake in older adults increases the risk of coronary heart disease [15,16]. Weeden et al. showed that more than 10% of older adults exceeded the UL intake for vitamin A, niacin, folic acid, and magnesium [16], and Sebastian et al. found that more than 10% of older men exceeded the UL for iron and zinc [15]. Our study showed fewer UL exceedances than other studies, probably because it evaluated people who used only one MVMS as a nutritional supplement. Thus, exceeding the ULs may be a concern when consuming multiple nutritional supplements. Nevertheless, the benefits of MVMS intake are clear, and the risk of side effects is not high, with some levels remaining below the recommended amount despite supplementation. These results highlight the importance of educating elderly individuals to consume mineral supplements without exceeding the UL of mineral or vitamin intake in food.

This study had some limitations. First, we used a 24 h recall method to evaluate nutrient intake; however, recall bias was possible due to memory errors. In addition, there may be differences in the accuracy of evaluating nutritional intake for 24 h recall only once. However, since this study involved an evaluation of nutrients in a population, not individuals, the difference was likely to be insignificant. Since only the limited nutrition data included in the survey were used, further analysis of vitamins B6, E, folic acid, magnesium, and zinc, which will not be analyzed in this study, is needed. This study was cross-sectional, and we only could suggest associations rather than causal relationships. The strengths of our study are that we used nationally representative KNHANES data from Korea, which provided helpful information regarding the vulnerable group and identified target groups for intervention. Although we did not conduct a direct multi-country comparison in this study, the results of previous studies conducted in various countries with nationally representative data and our results confirmed that older people were vulnerable to micronutrient intake [7,13,15,25,26]. This study was considered a cornerstone for conducting a pilot study to actively address inadequate nutritional intake for vulnerable older people in the future. Among those taking DS, those who took only a single MVMS were evaluated to more accurately evaluate the effect of MVMS consumption on nutritional intake in older Korean adults. In addition, more accurate information was obtained by matching the day of the 24 h recalls with the day of taking the MVMS. This is the first study to evaluate the micronutrient status of elderly Koreans at the national level by estimating micronutrient deficiency status and the improvement effect based on a single MVMS intake.

## 5. Conclusions

This study showed a cross-section of insufficient and excessive nutritional status through food intake and MVMS in older Korean adults. When Korean older adults consumed most micronutrients only with food, the intake of most nutrients was less than recommended. However, the intake of a single MVMS helped improve micronutrient intake in older Korean adults. Dietary supplement intake improved the nutritional status of vitamins and minerals, but the proportion of inadequate intake of some nutrients was still higher. Although excessive nutritional intake is not a cause for concern, the risk of excessive intake cannot be ruled out. Therefore, elderly Koreans should receive education on proper vitamin and mineral intake through a proper diet and, if necessary, MVMS.

## Figures and Tables

**Figure 1 nutrients-15-01561-f001:**
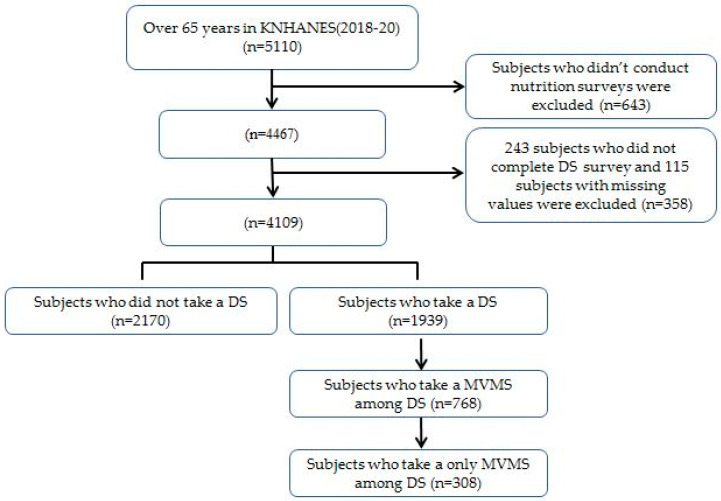
Flow chart of participations. Abbreviations: KNHANES, Korean National Nutrition and Health Survey; MVMS, multi-vitamin and mineral supplements; DS, dietary supplements.

**Figure 2 nutrients-15-01561-f002:**
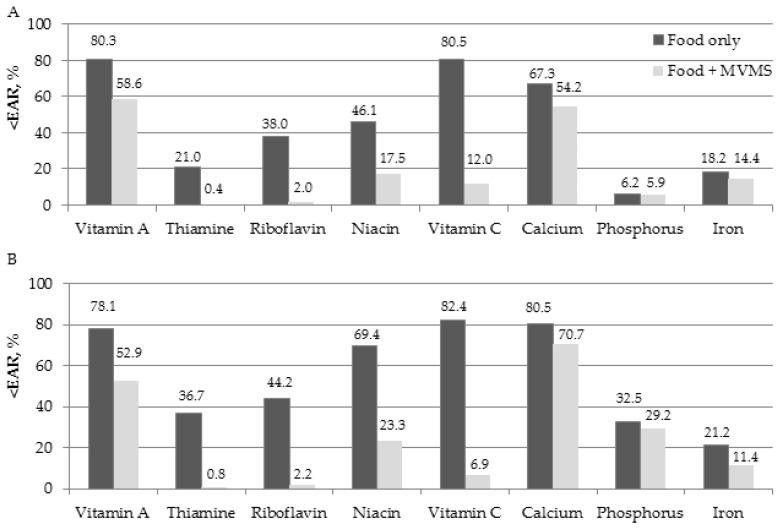
The percent of elderly whose intake was below EAR before and after taking MVMS in the MVMS user group; (**A**) males and (**B**) females. Abbreviations: EAR, estimated average requirement; MVMS, multi-vitamin and mineral supplements.

**Table 1 nutrients-15-01561-t001:** Baseline characteristics of participants in the food-only group and food + MVMS group.

Characteristics	Food-Only (*n* = 2170)	Food + MVMS (*n* = 308)	*p* Value
Age group			
65–74 years	53.5 (1.4)	61.2 (3.4)	0.046
75 years	46.5 (1.4)	38.8 (3.4)	
Sex, men	48.8 (1.1)	44.9 (3.2)	0.259
Education level			
Elementary school or lower	57.5 (1.5)	52.1 (3.7)	0.053
Graduation from middle school	16.9 (1.0)	15.6 (2.4)	
Graduation from high school	19.1 (1.0)	21.3 (2.7)	
University or more	6.5 (0.7)	11.1 (2.1)	
Household income quartile			
Low	49.6 (1.6)	40.2 (3.8)	0.048
Middle low	26.5 (1.4)	33.8 (3.5)	
Middle high	15.8 (1.1)	19.4 (2.8)	
High	8.0 (0.9)	6.6 (1.7)	
Smoking status			
Ex-smoker or non-smoker	89.3 (0.9)	95.2 (1.4)	0.003
Smoker	10.7 (0.9)	4.8 (1.4)	
Frequency of alcohol drinking			
None	51.9 (1.3)	50.7 (3.2)	0.471
1 time/week or less	30.9 (1.2)	34.1 (2.9)	
2–3 time/week	9.1 (0.7)	9.7 (2.0)	
4 time/week or more	8.1 (0.6)	5.5 (1.5)	
Physical activity			
lesser than mild	71.4 (1.4)	63.0 (3.3)	0.012
more than moderate	28.6 (1.4)	37.0 (3.3)	
Body mass index (kg/m^2^)			
Underweight (BMI < 18.5)	3.1 (0.5)	5.3 (1.3)	0.071
Normal (18.5 ≤ BMI < 23)	33.2 (1.2)	26.3 (3.0)	
Overweight (23 ≤ BMI < 25)	25.0 (1.1)	28.4 (2.9)	
Obesity (25 ≤ BMI)	38.7 (1.3)	40.1 (3.2)	
Have a disease	70.2 (1.1)	68.7 (2.9)	0.621
Have a cancer	9.1 (0.8)	6.7 (1.6)	0.210

Abbreviations: MVMS, multi-vitamin and mineral supplements; BMI, body mass index. Categorical variables were expressed as percentages and standard error (SE). Diseases included hypertension, hyperlipidemia, cardiovascular disease, stroke, and diabetes mellitus. Cancers included stomach, colon, liver, breast, cervical, lung, and other cancers.

**Table 2 nutrients-15-01561-t002:** Daily nutrient intake from food alone and included MVMS in males.

Nutrients	EAR	Food-Only (*n* = 991)	Food + MVMS (*n* = 139)			
		Food Alone	Food Alone	*p*-Value *	Food Include MVMS	*p*-Value ^†^
Total calories (kcal/day)	NA	1750.85 ± 21.81	1886.62 ± 55.00	0.021	NA	NA
Carbohydrate (g/day)	NA	289.47 ± 3.36	302.19 ± 9.85	0.227	NA	NA
Protein (g/day)	NA	59.55 ± 0.91	63.72 ± 2.76	0.152	NA	NA
Fat (g/day)	NA	29.77 ± 0.77	34.49 ± 2.29	0.047	NA	NA
Fiber (g/day)	NA	25.95 ± 0.51	27.06 ± 1.22	0.401	NA	NA
Vitamin A (μg/day)	510 (65–74 years) 500 (≥75 years)	309.90 ± 10.08	420.87 ± 104.06	0.294	702.73 ± 111.49	<0.001
Thiamine (mg/day)	0.9	1.23 ± 0.02	1.26 ± 0.04	0.400	34.28 ± 2.56	<0.001
Riboflavin (mg/day)	1.2 (65–74 years) 1.1 (≥75 years)	1.27 ± 0.02	1.45 ± 0.07	0.031	20.21 ± 2.71	<0.001
Niacin (mg/day)	11 (65–74 years) 10 (≥75 years)	11.22 ± 0.20	12.23 ± 0.65	0.142	36.53 ± 2.80	<0.001
Vitamin C (mg/day)	75	53.73 ± 2.03	60.44 ± 6.74	0.336	200.72 ± 18.77	<0.001
Calcium (mg/day)	600	477.15 ± 11.39	557.31 ± 37.28	0.038	630.38 ± 39.27	<0.001
Phosphorus (mg/day)	580	957.35 ± 14.40	1039.02 ± 41.16	0.065	1050.12 ± 41.05	0.036
Iron (mg/day)	7	11.68 ± 0.26	12.60 ± 0.67	0.200	14.95 ± 0.90	<0.001

Abbreviations: MVMS, multi-vitamin and mineral supplements; EAR, estimated average requirement; NA, not available. Data are shown as estimated means ± standard error. Multivariate regression adjusted for age, sex, education level, household incomes, smoking status, frequency of drinking, physical activity, BMI, and presence of diseases and cancer. * Comparisons of nutrient intakes from food alone between food-only group and “food + MVMS group”. ^†^ Comparisons of total nutrient intakes between food-only group and food + MVMS group.

**Table 3 nutrients-15-01561-t003:** Daily nutrient intake from food alone and included MVMS in females.

Nutrients	EAR	Food-Only (*n* = 1179)	Food + MVMS (*n* = 169)			
		Food Alone	Food Alone	*p*-Value *	Food Include MVMS	*p*-Value ^†^
Total calories (kcal/day)	NA	1346.96 ± 20.46	1342.32 ± 44.19	0.926	NA	NA
Carbohydrate (g/day)	NA	241.01 ± 3.63	227.32 ± 7.46	0.109	NA	NA
Protein (g/day)	NA	44.02 ± 0.87	46.85 ± 1.95	0.205	NA	NA
Fat (g/day)	NA	21.43 ± 0.70	25.91 ± 1.91	0.025	NA	NA
Fiber (g/day)	NA	21.32 ± 0.43	20.76 ± 1.04	0.624	NA	NA
Vitamin A (μg/day)	410	260.91 ± 14.99	259.96 ± 25.35	0.973	621.69 ± 62.20	<0.001
Thiamine (mg/day)	0.8 (65–74 years) 0.7 (≥75 years)	0.97 ± 0.02	0.90 ± 0.03	0.080	31.89 ± 2.40	<0.001
Riboflavin (mg/day)	0.9 (65–74 years) 0.8 (≥75 years)	0.95 ± 0.02	1.02 ± 0.05	0.170	19.44 ± 2.36	<0.001
Niacin (mg/day)	10 (65–74 years) 9 (≥75 years)	8.16 ± 0.17	8.04 ± 0.36	0.778	31.05 ± 2.25	<0.001
Vitamin C (mg/day)	75	50.82 ± 2.26	46.96 ± 4.97	0.486	197.92 ± 13.06	<0.001
Calcium (mg/day)	600	372.77 ± 8.83	382.28 ± 20.99	0.673	464.67 ± 25.52	<0.001
Phosphorus (mg/day)	580	725.31 ± 13.06	753.78 ± 30.54	0.401	762.76 ± 30.45	0.268
Iron (mg/day)	6 (65–74 years) 5 (≥75 years)	8.65 ± 0.17	8.77 ± 0.34	0.760	13.15 ± 0.92	<0.001

Abbreviations: MVMS, multi-vitamin and mineral supplements; EAR, estimated average requirement; NA, not available. Data are shown as estimated means ± standard error. Multivariate regression adjusted for age, sex, education level, household incomes, smoking status, frequency of drinking, physical activity, BMI, and presence of diseases and cancer. * Comparisons of nutrient intakes from food alone between food-only group and “food + MVMS group”. ^†^ Comparisons of total nutrient intakes between food-only group and ‘food + MVMS’ group.

**Table 4 nutrients-15-01561-t004:** Changed percentage of micronutrient intake exceeding upper limit after eating a single MVMS.

Nutrients	Male			Female		
	Food-Only	Food + MVMS	Change (%)	Food-Only	Food + MVMS	Change (%)
Vitamin A	1.0 (1.0)	1.5 (1.1)	0.5	0	1.4 (1.0)	1.4
Vitamin C	0	0	0	0	0	0
Calcium	0	0	0	0	0	0
Phosphorus	0	0	0	0	0	0
Iron	1.0 (1.0)	2.1 (1.5)	1.1	0	1.2 (1.2)	1.2

Abbreviations: MVMS, multi-vitamin and mineral supplements. Categorical variables were expressed as percentages and standard error (SE). Change (%) was calculated by subtracting the percentage of food only group from the “food + MVMS group”.

## Data Availability

The data presented in this study are available on reasonable request from the corresponding author.

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
