# Peer review of "Effect of a Single Multi-Vitamin and Mineral Supplement on Nutritional Intake in Korean Elderly: Korean National Health and Nutrition Examination Survey 2018–2020"

_nutrients, 2023, doi:10.3390/nu15071561_

Round 1

Reviewer 1 Report

1. Wouldn't it be better to make a multinational comparison? A domestic report as a pilot study would be acceptable.

2. The question remains as to why the types of vitamins and minerals were limited this time. did you do a screening?

3. P-values are not shown in the results despite statistical processing. how was it actually?

4. It would be better to use a graph or the like.

Author Response

I really appreciate your thorough review on our study.

  1. Wouldn't it be better to make a multinational comparison? A domestic report as a pilot study would be acceptable.

Answer1) Although we checked similar studies from several countries as references, we were not doing this study to make multi-national comparisons. This study was designed and conducted to evaluate the nutritional status of micronutrient intake and the improvement effect of multi-vitamin and mineral supplements (MVMS) intake in elderly Koreans. In the following thesis, we will develop new contents that reflect your opinions.

  1. The question remains as to why the types of vitamins and minerals were limited this time. did you do a screening?

Answer 2) This study aimed to improve micronutrient intake; therefore, among dietary supplements that had the most impact, MVMS formulations were selected and analyzed. Data from the Korean National Health and Nutrition Examination Survey include MVMS formulations containing overall vitamins and minerals and other formulations containing high amounts of specific nutrients. Therefore, we selected and analyzed only multivitamin and mineral formulations to further reduce nutrient-specific errors that could lead to over-supplementation and to evaluate the effectiveness of relatively consistent vitamin and mineral formulations.

  1. P-values are not shown in the results despite statistical processing. how was it actually?

Answer3) It was found that there was some error in presenting the table, and the P-value was presented in tables 2 and 3. In figure 2 and table 4, only differences were shown, so P-values were not presented.

  1. It would be better to use a graph or the like.

Answer 4) Following kind advice, we presented Table 4 as a graph and changed it to figure 2, which made it much easier to understand visually. Additionally, Table 4 was changed to a supplementary table (Table S1) to inform the difference value of the change in the below EAR ratio.

Figure 2. The percent of elderly whose intake was below EAR before and after taking MVMS in the MVMS user group; (A) males and (B) females. Abbreviation: EAR, estimated average requirement; MVMS, multi-vitamin and mineral supplements.

Reviewer 2 Report

1. "BMI was classified as the normal range < 23 kg/m2" - The lower limit of the norm established in this way may also include underweight people.

2. Where are the Tables?

Maybe I'm tired, but not being able to have full text with tables makes it hard for me to write a review.

Author Response

I really appreciate your thorough review on our study

We have enclosed an English proofreading certificate.

  1. "BMI was classified as the normal range < 23 kg/m2" - The lower limit of the norm established in this way may also include underweight people.

Answer1) As reviewers mentioned, a few subjects were included in the underweight group. We reclassified the BMI group and analyzed it again by classifying it into 4 BMI groups underweight, normal, overweight, and obese). With the application of the newly applied BMI group, there was a small change in the result value, but no change in statistical results. Changed results due to the changed BMI definition were additionally marked in red in the table 1, 2, 3 and S1.

Body mass index (kg/m2)

Underweight (BMI<18.5)

3.1 (0.5)

5.3 (1.3)

0.071

Normal (18.5≤BMI<23)

33.2 (1.2)

26.3 (3.0)

Overweight (23≤BMI<25)

25.0 (1.1)

28.4 (2.9)

Obesity (25≤BMI)

38.7 (1.3)

40.1 (3.2)

  1. Where are the Tables?

Answer2) Sorry for the inconvenience to read. We provided a table attached separately, but it seemed hard to read. In the process of writing this time, tables were inserted into the text.

Reviewer 3 Report

Thank you for the opportunity to review this manuscript. The topic is interesting, but I have suggestions for improving this manuscript.

At first, the introduction could be expanded with other information, especially regarding the choice of micronutrients to be supplemented. In addition, it's necessary to explicitly the role of these micronutrients in the physiologic function. Here are some recent references "Sinopoli A, Caminada S, Isonne C, Santoro MM, Baccolini V. What Are the Effects of Vitamin A Oral Supplementation in the Prevention and Management of Viral Infections? A Systematic Review of Randomized Clinical Trials. Nutrients. 2022 Oct 1;14(19):4081. doi: 10.3390/nu14194081. PMID: 36235733; PMCID: PMC9572963.", Kumar, R. R., Singh, L., Thakur, A., Singh, S., & Kumar, B. (2022). Role of Vitamins in Neurodegenerative Diseases: A Review. CNS & Neurological Disorders-Drug Targets (Formerly Current Drug Targets-CNS & Neurological Disorders)21(9), 766-773.  I suggest better underlining the aim of this study.

In the methods, it's indispensable to create a table with baseline characteristics of the population. In addition, it's necessary to have a table with multivariate regression analysis.

Author Response

I really appreciate your thorough review on our study.

  1. At first, the introduction could be expanded with other information, especially regarding the choice of micronutrients to be supplemented. In addition, it's necessary to explicitly the role of these micronutrients in the physiologic function. Here are some recent references "Sinopoli A, Caminada S, Isonne C, Santoro MM, Baccolini V. What Are the Effects of Vitamin A Oral Supplementation in the Prevention and Management of Viral Infections? A Systematic Review of Randomized Clinical Trials. Nutrients. 2022 Oct 1;14(19):4081. doi: 10.3390/nu14194081. PMID: 36235733; PMCID: PMC9572963.", Kumar, R. R., Singh, L., Thakur, A., Singh, S., & Kumar, B. (2022). Role of Vitamins in Neurodegenerative Diseases: A Review. CNS & Neurological Disorders-Drug Targets (Formerly Current Drug Targets-CNS & Neurological Disorders)21(9), 766-773.  I suggest better underlining the aim of this study.

Answer1) In the introduction, we added information to clarify the choice of micronutrients to be supplemented and explicitly the role of these micronutrients in the physiologic function. We also tried to make the aim of this study clearer.

Micronutrient deficiencies can adversely affect various aspects of the health of elderly individuals, including immune function [8,9], frailty [10], osteoporosis [11], and longevity [12] through multiple pathways related to cells differentiation, oxidative stress, muscle and bone metabolism, inflammation, and decreased immunity [10,13,14]. Recent studies have also suggested that micronutrient deficiencies, such as vitamins, can cause abnormal brain functions such as oxidative stress, mitochondrial dysfunction, and neurodegeneration, leading to various neurological disorders such as Alzheimer's disease, Parkinson's disease, and depression [14]. An increase in dementia or other neurological diseases can be another burden in an aging population.

Some studies conducted in Korea just showed the DS usage patterns and micronutrient deficiencies in elderly Korean individuals [23], and few studies suggested the improvement effect after taking multi-vitamin and mineral supplements (MVMS) in older people. Therefore, the improvements in older Korean individuals taking MVMS with food need to be clarified. 

We hypothesized that the intakes of micronutrients in older Koreans need to be increased to meet the recommended nutritional intake and that the MVMS may help improve nutritional status. In order to exclude overlapping effects due to the combination with other DS and to determine the effect of only a single MVMS, this study was to be conducted with subjects who take only MVMS purely by excluding all multi-users. To address these hypotheses, in this study, based on data from the large nationally representative Korean National Nutrition and Health Survey (KNHANES) 2018–2020, we investigated the micronutrient intake status in older Korean people.

  1. In the methods, it's indispensable to create a table with baseline characteristics of the population. In addition, it's necessary to have a table with multivariate regression analysis.

Answer 2) It was found that there was some error in presenting the table of the first manuscript, the contents related to multivariate correction variables were not written in tables 2 and 3. However, in table 4, the related contents were properly corrected in tables 2 and 3.

Round 2

Reviewer 2 Report

Thank you for the additions and the inclusion of tables in the text. The publication in this form is interesting and gives an idea of the value of possible supplementation among the elderly. Unfortunately, we do not know what long-term effects this could have and how it will improve the quality of life of these people. Nevertheless, such cross-sectional research is valuable.

Reviewer 3 Report

The manuscript is ok in this version.